# Duration of type 2 diabetes and remission rates after bariatric surgery in Sweden 2007–2015: A registry-based cohort study

Anders Jans[1], Ingmar Näslund[1], Johan Ottosson[1], Eva Szabo[1], Erik Näslund[2], Erik Stenberg[1]*

**1** Department of Surgery, Faculty of Medicine and Health, Örebro University, Örebro, Sweden, **2** Division of Surgery, Department of Clinical Sciences, Danderyd Hospital, Karolinska Institutet, Stockholm, Sweden

* erik.stenberg@regionorebrolan.se

## Abstract

### Background

Although bariatric surgery is an effective treatment for type 2 diabetes (T2D) in patients with morbid obesity, further studies are needed to evaluate factors influencing the chance of achieving diabetes remission. The objective of the present study was to investigate the association between T2D duration and the chance of achieving remission of T2D after bariatric surgery.

### Methods and findings

We conducted a nationwide register-based cohort study including all adult patients with T2D and BMI $\geq$ 35 kg/m$^2$ who received primary bariatric surgery in Sweden between 2007 and 2015 identified through the Scandinavian Obesity Surgery Registry. The main outcome was remission of T2D, defined as being free from diabetes medication or as complete remission (HbA1c < 42 mmol/mol without medication). In all, 8,546 patients with T2D were included. Mean age was 47.8 ± 10.1 years, mean BMI was 42.2 ± 5.8 kg/m$^2$, 5,277 (61.7%) were women, and mean HbA1c was 58.9 ± 17.4 mmol/mol. The proportion of patients free from diabetes medication 2 years after surgery was 76.6% (n = 6,499), and 69.9% at 5 years (n = 3,765). The chance of being free from T2D medication was less in patients with longer preoperative duration of diabetes both at 2 years (odds ratio [OR] 0.80/year, 95% CI 0.79–0.81, p < 0.001) and 5 years after surgery (OR 0.76/year, 95% CI 0.75–0.78, p < 0.001). Complete remission of T2D was achieved in 58.2% (n = 2,090) at 2 years, and 46.6% at 5 years (n = 681). The chance of achieving complete remission correlated negatively with the duration of diabetes (adjusted OR 0.87/year, 95% CI 0.85–0.89, p < 0.001), insulin treatment (adjusted OR 0.25, 95% CI 0.20–0.31, p < 0.001), age (adjusted OR 0.94/year, 95% CI 0.93–0.95, p < 0.001), and HbA1c at baseline (adjusted OR 0.98/mmol/mol, 95% CI 0.97–0.98, p < 0.001), but was greater among males (adjusted OR 1.57, 95% CI 1.29–1.90, p < 0.001) and patients with higher BMI at baseline (adjusted OR 1.07/kg/m$^2$, 95% CI 1.05–1.09, p < 0.001). The

**Data Availability Statement:** Data cannot be shared publicly because of patient confidentiality under current Swedish legislation. Data are available from the Scandinavian Obesity Surgery

Registry (contact via www.ucr.uu.se/soreg/) for researchers who meet the criteria for access to confidential data.

**Funding:** This work was supported by grants from the Örebro Region County Council (AJ, grant number: OLL-915571; ESt grant number:OLL-884791), the Bengt Ihre Foundation (ESt), Stockholm County Council (EN), SRP Diabetes (EN) and the NovoNordisk Foundation (EN). The funders had no role in study design, data Collection and alaysis, decision to publish, or preparation of the manuscript

**Competing interests:** I have read the journal's policy and the authors of this manuscript have the following competing interests: IN has received consultant fees from Baricol Bariatrics AB, Sweden and Ethicon Endosurgery, Johnson & Johnson for work unrelated to the context of the present study. JO has received consultant fees from Vifor Pharma AB, and Ethicon Endosurgery, Johnson & Johnson for work unrelated to the context of the present study. None of the remaining authors declares any conflict of interest.

**Abbreviations:** %EBMIL, percentage excess BMI loss; %TWL, percentage total weight loss; OR, odds ratio; SOReg, Scandinavian Obesity Surgery Registry; T2D, type 2 diabetes.

main limitations of the study lie in its retrospective nature and the low availability of HbA1c values at long-term follow-up.

## Conclusions

In this study, we found that remission of T2D after bariatric surgery was inversely associated with duration of diabetes and was highest among patients with recent onset and those without insulin treatment.

## Author summary

### Why was this study done?

- Bariatric surgery is an effective treatment for type 2 diabetes in obese patients.

- Previous studies have identified several factors affecting the chance of diabetes remission after bariatric surgery, such as age, HbA1c, insulin therapy, and diabetes duration.

- The main purpose of the study was to analyze the relationship between preoperative diabetes duration and the likelihood of achieving diabetes remission after bariatric surgery in a large nationwide patient population.

### What did the researchers do and find?

- The registry-based nationwide study included 8,546 patients with type 2 diabetes who underwent bariatric surgery in Sweden between 2007 and 2015.

- In total, complete diabetes remission was achieved in 58.2% of patients after 2 years and 46.6% after 5 years.

- In this study, we found that remission of type 2 diabetes after bariatric surgery was inversely associated with duration of diabetes and was highest among patients with recent onset and those without insulin treatment.

### What do these findings mean?

- The relationship between preoperative diabetes duration and chance of diabetes remission is valuable in analyzing the potential benefit compared to risk related to bariatric surgery, and can therefore be used to prioritize for surgery those patients who are most likely to achieve diabetes remission.

## Introduction

Since 1980, the average body mass index (BMI) has risen globally, yielding a prevalence of obesity as high as 10.8% in men and 14.9% in women [1,2]. In several countries, the prevalence may approach or even reach beyond 50% of the adult population [3]. Obesity is a strong risk factor for developing type 2 diabetes (T2D) [4]. Consequently, T2D contributes greatly to the overall burden of disease [5]. Compared to nonsurgical treatment, bariatric surgery provides a more effective way to achieve long-term weight loss in obese individuals [6], and increases overall survival in this patient group [6]. Obese patients with T2D have a good chance of achieving long-term T2D remission after surgery [7–11], although some patients who initially remit later relapse [7,12]. The duration of diabetes prior to surgery, glycemic control, insulin use, age, and postoperative weight loss have previously been suggested as factors influencing the chance of achieving remission of T2D after bariatric surgery [9,11–16]. These studies generally comprise small samples, some over rather short periods of time. Larger studies are thus needed to ascertain which factors are associated with diabetes remission.

The aim of the present study was to evaluate the impact of preoperative duration of diabetes and other factors on the chance of achieving diabetes remission after bariatric surgery.

## Methods

This study was a retrospective cohort study on prospectively collected data from the Scandinavian Obesity Surgery Registry (SOReg). The registry was launched in 2007 and covers virtually all bariatric surgical procedures in Sweden, with very high registration validity [17]. All patients who received primary gastric bypass or sleeve gastrectomy surgery between January 1, 2007, and December 31, 2015, and registered in SOReg were considered for inclusion in the study. Only patients with T2D as defined by the American Diabetes Association were included [4]. Although an original study plan was decided on by the authors, it was not documented beforehand. After the study began, percentage total weight loss (%TWL) was changed to percentage excess BMI loss (%EBMIL) in the multivariable analysis as a response to review; no other changes were made to the original plan.

By using the Swedish personal identification number, unique to each citizen, the SOReg data file was linked to the Swedish National Patient Register, the Swedish Population Register (for mortality data), the Swedish Prescribed Drug Register, and Statistics Sweden. Information on baseline characteristics, surgery, and follow-up were based on data from SOReg. Since cardiovascular comorbidity and previous pulmonary embolus/deep venous thrombosis are not obligatory variables in SOReg, data on these conditions were based on combined data from the Swedish National Patient Register and SOReg. Preoperative duration of diabetes was based on a combination of data from SOReg, the National Patient Registries and the Swedish Prescribed Drug Register. Data on specific pharmaceutical treatment for diabetes were based on data from SOReg and the Swedish Prescribed Drug Register. Information on education was based on patient-specific data from Statistics Sweden.

### Procedures

The surgical method for gastric bypass is highly standardized in Sweden, with 99% being an antecolic/antegastric gastric bypass procedure with a small gastric pouch [18]. The sleeve gastrectomy is less standardized, but routinely performed using a 32–36 Fr bougie with the gastric division starting 5 cm from the pylorus and ending 1 cm from the angle of His. Perioperative care closely follows the Enhanced Recovery After Surgery guidelines, with early mobilization, routine thromboprophylaxis, and start of oral fluids on the day of surgery [19].

## Outcomes and definitions

The main outcome was remission of diabetes 2 and 5 years after surgery. To allow comparison with previous studies, this was defined both as being without medication at these points in time (including a time frame of ±6 months) and achieving complete remission of diabetes.

Complete remission of diabetes was defined as a glycosylated hemoglobin A1c (HbA1c) < 42 mmol/mol (6.0%) without medical treatment. Partial remission of diabetes was defined as HbA1c of 42–48 mmol/mol (6.0%–6.5%) without medical treatment, in accordance with the recommendations of the American Society for Metabolic and Bariatric Surgery [20]. Controlled diabetes was defined as HbA1c < 48 mmol/mol (<6.5%) with medical treatment.

Comorbidity was defined as a medical condition requiring pharmaceutical treatment, or continuous positive airway pressure treatment in the case of sleep apnea. Cardiovascular comorbidity was defined as a history of ischemic heart disease, angina pectoris, cardiomyopathy, cardiac failure, or arrhythmic heart disease.

Postoperative weight loss was presented as change in BMI ($\Delta$BMI = initial BMI − postoperative BMI), percentage total weight loss (%TWL = 100 × weight loss/preoperative weight), and percentage excess BMI loss (%EBMIL = 100 × [initial BMI − postoperative BMI]/[initial BMI − 25]). Postoperative complications were classified according to the Clavien–Dindo classification, and having a postoperative complication was defined as Clavien–Dindo $\geq$ 1. Postoperative complications graded $\geq$3b were considered serious complications [21]. Since more specific classification of postoperative complications was made obligatory in SOReg starting January 1, 2010, only patients operated on from this date onwards were included in the analysis of serious postoperative complications.

## Statistics

Baseline data (before surgery) and follow-up data are presented as numbers of individuals (*n*) with percentages of patients for categorical values, mean ± standard deviation (SD) for continuous variables assuming normal distribution, and median ± interquartile range (IQR) for continuous variables not assuming normal distribution. Univariable logistic regression analyses were conducted to evaluate risks related to the major endpoints of the study. The correlation between diabetes duration and remission of diabetes was evaluated using the Spearman correlation test. Based on previous studies and plausible impact of preoperatively available factors, diabetes duration, insulin treatment, BMI, age, sex, HbA1c, education, %EBMIL, and surgical method were incorporated in the multivariable logistic regression analyses [22,23]. Missing data were handled by listwise deletion. $p < 0.05$ was considered to represent statistical significance.

SPSS Statistics version 25 (IBM, Armonk, New York, US) was used for all analyses.

## Ethics

The study was approved by the Regional Ethics Committee in Stockholm (Dnr 2013/535-31/5, 2014/1639-32, and 2017/857-32) and conducted in accordance with the ethical standards of the 1964 Helsinki Declaration and its later amendments. No written consent was obtained from the study participants. However, in accordance with Swedish legislation, all participants were informed of the research and quality registry and that the data would be used in clinical research, giving the patients the right to decline participation.

## Results

During the inclusion period, 8,546 patients with T2D according to the definition of the American Diabetes Association were identified in SOReg. Two years after surgery, 57 patients had died, leaving 8,489 (99%) patients available for analysis of pharmaceutical usage. After exclusion of patients who died prior to the 5-year follow-up ($n$ = 178) and patients who did not reach this time point by the end of the study ($n$ = 2,980), 5,388 patients were available for analyses on pharmaceutical use at 5 years. HbA1c was available at 2 years for 3,594 patients (42% of patients reaching the 2-year follow-up) and at 5 years for 1,460 patients (27% of patients reaching the 5-year follow-up). Follow-up data on day 30 were available for 8,448 patients (99% of all patients) (Fig 1). Data on baseline characteristics for the study group are presented in Table 1.

### Surgical data

A total of 8,112 patients underwent a gastric bypass procedure (94.9%), and 434 a sleeve gastrectomy (5.1%). A laparoscopic procedure was completed in 8,171 operations (95.6%), 111 operations were converted to open surgery (1.3%), and a primary open procedure was performed in 264 operations (3.1%). The median hospital stay was 2 days (IQR 1–2 days). A postoperative complication (Clavien–Dindo $\geq$ 1) occurred within 30 days after 863 operations (10.2%). A serious postoperative complication occurred within 30 days after 241 operations (3.5%). Mean BMI loss at 2 years was 11.9 ± 4.6 kg/m$^2$, %TWL 28.1% ± 9.2%, and %EBMIL 72.7% ± 24.7%. Mean BMI loss at 5 years was 10.7 ± 4.7 kg/m$^2$, %TWL 25.2% ± 9.6%, and %EBMIL 64.4% ± 25.6%.

### Impact on diabetes medication

At baseline, 4,192 patients (49.1%) received oral medical treatment alone for diabetes, 453 received insulin alone (5.3%), 14 received a GLP-1 analogue alone (0.2%), and 1,973 received a

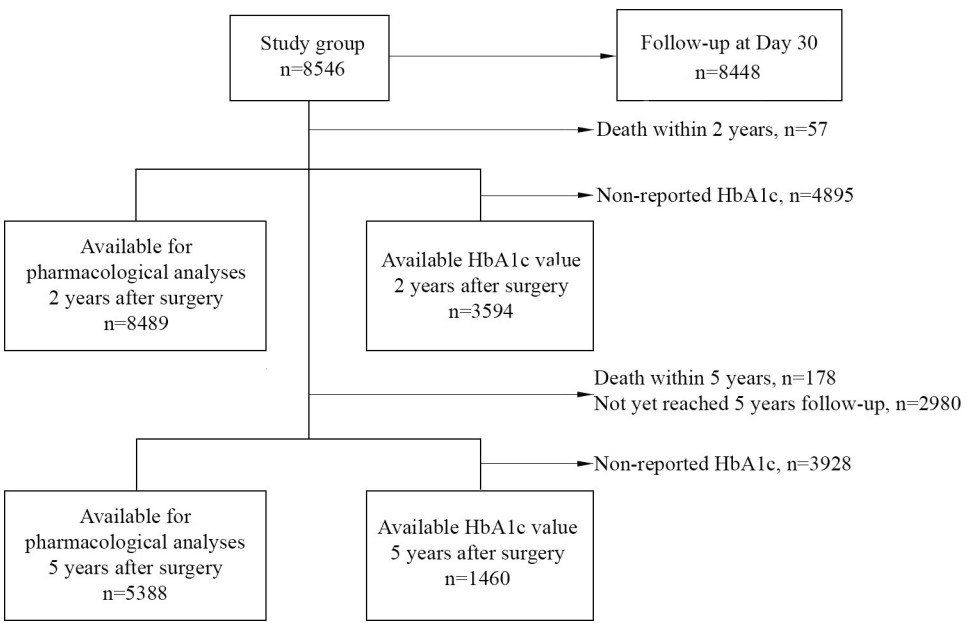

**Fig 1. Study flowchart describing availability for analyses.**

**Table 1. Baseline characteristics.**

| Characteristic | Missing data, *n* (%) | Mean ± SD or *n* (%) |
|---|---|---|
| BMI, kg/m$^2$ | 0 (0.0%) | 42.2 ± 5.8 |
| Age, years | 0 (0.0%) | 47.8 ± 10.1 |
| Sex | 0 (0.0%) | |
| Female | | 5,277 (61.7%) |
| Male | | 3,269 (38.3%) |
| Comorbidity | | |
| Sleep apnea | 0 (0.0%) | 1,607 (18.8%) |
| Cardiovascular comorbidity | 0 (0.0%) | 972 (11.4%) |
| Hypertension | 0 (0.0%) | 4,810 (56.3%) |
| Dyslipidemia | 0 (0.0%) | 2,679 (31.3%) |
| Dyspepsia/gastroesophageal reflux disease | 0 (0.0%) | 1,087 (12.7%) |
| Depression | 0 (0.0%) | 1,405 (16.4%) |
| Previous pulmonary embolus/deep venous thrombosis | 0 (0.0%) | 219 (2.6%) |
| Glycosylated hemoglobin A1c, mmol/mol | 1,134 (13.3%) | 58.9 ± 17.4 |
| Education | 59 (0.7%) | |
| Primary education (≤9 years) | | 1,691 (19.9%) |
| Secondary education (10–12 years) | | 5,027 (59.2%) |
| Higher education ≤3 years | | 898 (10.6%) |
| Higher education >3 years | | 871 (10.3%) |

combination of oral treatment and insulin (23.1%). The remaining 1,914 patients received non-pharmacological management only for their diabetes (22.5%).

Two years after surgery, 1,990 patients (23.4%) received pharmacological treatment for diabetes: 1,328 received oral medical treatment alone for diabetes (15.6%), 228 received insulin alone (2.7%), 19 received a GLP-1 analogue alone (0.2%), and 415 received a combination of oral treatment and insulin (4.9%). The remaining 6,499 did not receive medical treatment for diabetes 2 years after surgery (76.6%). The chance of being free of diabetes medication was less with longer preoperative duration of diabetes (odds ratio [OR] 0.80/year, 95% CI 0.79–0.81, $p$ < 0.001; Spearman coefficient −0.43, $p$ < 0.001) (Table 2 and Fig 2).

**Table 2. Numbers free from medical treatment 2 and 5 years after surgery.**

| Diabetes duration, years | *n* (%) free from medical treatment at follow-up | |
|---|---|---|
| | 2 years | 5 years |
| <1 | 2,473 (96.1%) | 1,535 (94.2%) |
| 1 | 899 (86.7%) | 553 (83.0%) |
| 2 | 674 (82.9%) | 404 (75.0%) |
| 3 | 549 (75.6%) | 322 (66.1%) |
| 4 | 437 (74.3%) | 252 (62.6%) |
| 5 | 407 (69.3%) | 252 (59.0%) |
| 6–7 | 469 (58.0%) | 223 (43.0%) |
| 8–9 | 270 (51.5%) | 95 (34.5%) |
| 10–12 | 187 (44.0%) | 76 (32.1%) |
| 13–15 | 61 (34.3%) | 29 (33.0%) |
| 16–20 | 49 (35.8%) | 17 (22.1%) |
| 21–25 | 21 (32.3%) | 6 (17.1%) |
| ≥26 | 3 (10.7%) | 1 (7.7%) |

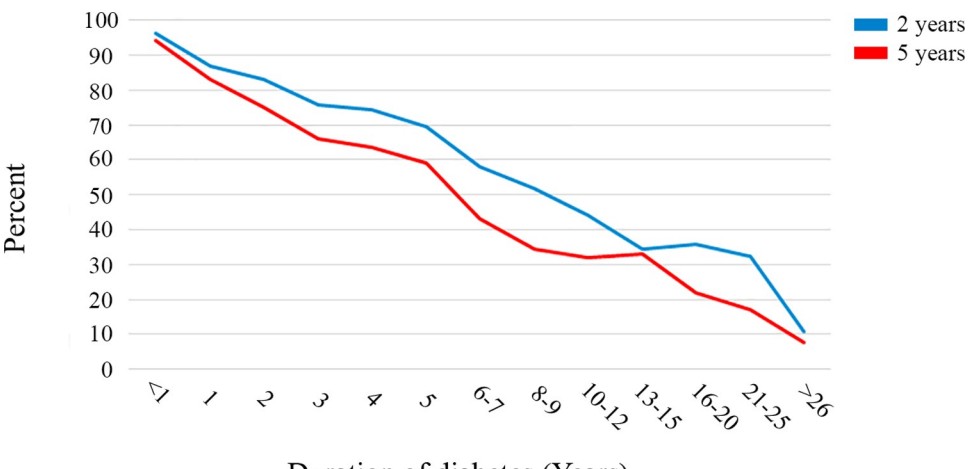

**Fig 2. Proportion of patients free from medication at 2 and 5 years after surgery.**

Five years after surgery, 1,623 patients (30.1%) received pharmacological treatment for diabetes: 1,095 received oral medical treatment alone for diabetes (20.3%), 152 received insulin alone (3.0%), 24 received a GLP-1 analogue alone (0.4%), and 342 received a combination of oral treatment and insulin (6.4%). The remaining 3,765 did not receive medical treatment for diabetes 5 years after surgery (69.9%). As at 2 years, the chance of being free of diabetes medication 5 years after surgery was less with longer preoperative duration of diabetes (OR 0.76/year, 95% CI 0.75–0.78, $p < 0.001$; Spearman coefficient −0.48, $p < 0.001$) (Table 2 and Fig 2).

## Diabetes remission

Two years after surgery, 2,090 patients had complete remission of their diabetes (58.2%), 429 patients achieved partial remission (11.9%), 428 had controlled diabetes on medication (11.9%), and 647 patients (7.6%) still had a HbA1c $\geq$ 48 mmol/mol on pharmacological treatment (Fig 3). The chance of achieving complete remission was less the longer the preoperative duration of diabetes (OR 0.78/year, 95% CI 0.76–0.79, $p < 0.001$; Spearman coefficient −0.46, $p < 0.001$).

Five years after surgery, 681 patients had complete remission of their diabetes (46.6%), 175 patients had partial remission (12.0%), 188 had controlled diabetes on medication (12.9%), and 416 patients (28.5%) still had a HbA1c > 48 mmol/mol on pharmacological treatment (Fig 4). The chance of achieving complete remission was still less the longer the preoperative duration of diabetes (OR 0.77/year, 95% CI 0.74–0.80, $p < 0.001$; Spearman coefficient −0.44, $p < 0.001$).

## Multivariable analysis

Of the selected preoperative factors potentially influencing diabetes remission, longer duration of diabetes, higher baseline HbA1c, high age, and insulin treatment were all associated with lower remission rates 2 years after surgery. Higher BMI, higher excess BMI loss, and male sex in patients with T2D were associated with higher remission rates. Higher education and gastric bypass were associated with better chance of achieving diabetes remission in the univariable analyses, but the associations did not remain statistically significant in the multivariable analyses (Table 3).

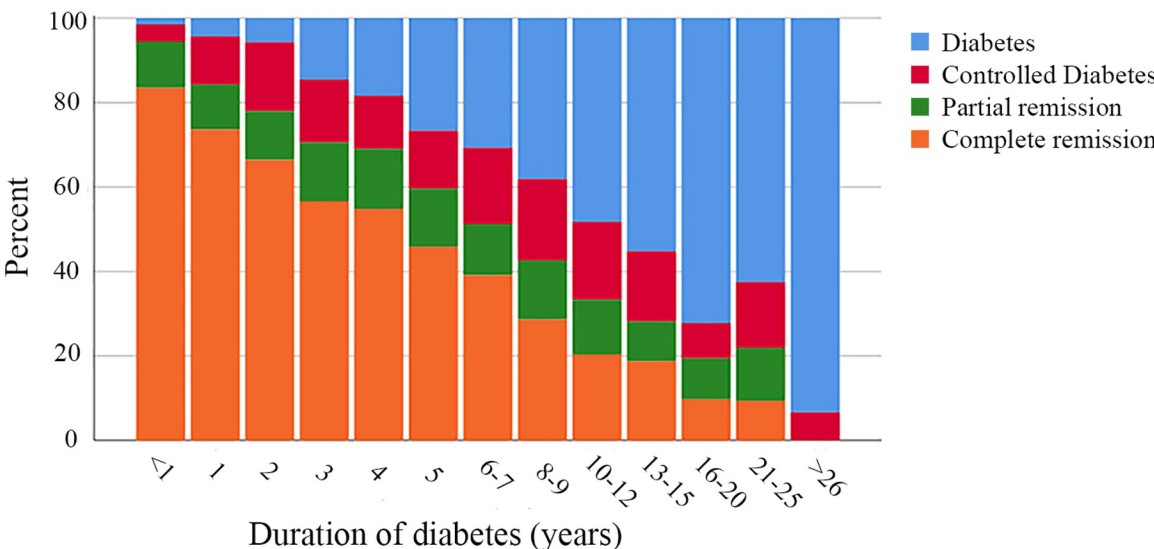

**Fig 3. Stacked histogram of 2-year remission in relation to duration of diabetes.**

## Discussion

In the present study, a strong negative correlation between preoperative duration of T2D and the chance of diabetes remission after bariatric surgery was seen. The negative correlation was linear even over very long diabetes duration, supporting the results of previous studies [9,11–16]. Other factors of importance were insulin treatment and baseline HbA1c as markers of severity of disease, age, sex, and postoperative weight loss.

Remission rate depends on the definition used to define diabetes remission. With the current definition of complete remission, the remission rate in the study cohort was comparable to that reported in previous studies [9,13]. Thus, our results support the already established position of bariatric surgery as an important tool in the treatment of T2D in patients with morbid obesity [10]. However, the benefits of bariatric surgery appear to be greatest amongst patients with more recent onset of T2D and those with less severe disease.

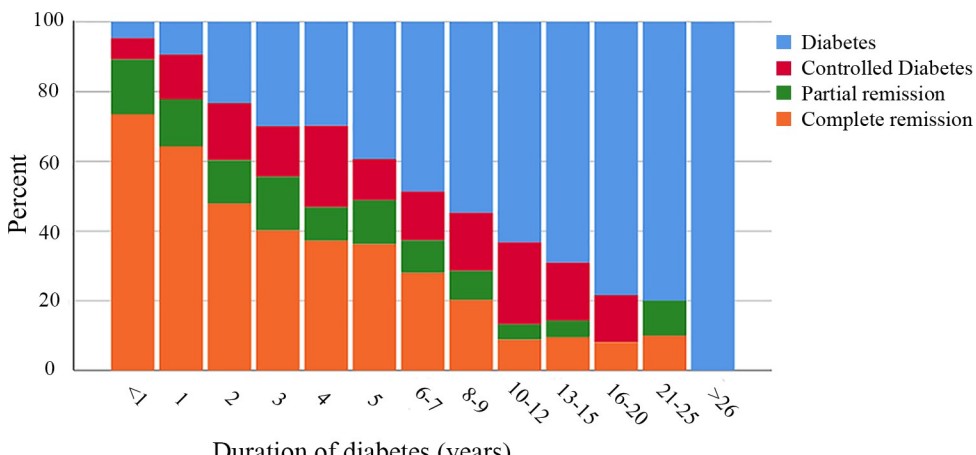

**Fig 4. Stacked histogram of 5-year remission in relation to duration of diabetes.**

**Table 3. Odds ratios (ORs) for reaching complete diabetes remission 2 years after surgery.**

| Characteristic | Unadjusted OR (95% CI) | Adjusted OR (95% CI)[1] | Adjusted p-Value[1] |
|---|---|---|---|
| Preoperative diabetes duration | 0.78 (0.76–0.79) | 0.87 (0.85–0.89) | <0.001 |
| Baseline HbA1c | 0.96 (0.95–0.97) | 0.98 (0.97–0.98) | <0.001 |
| Insulin treatment at baseline | 0.15 (0.12–0.16) | 0.25 (0.20–0.31) | <0.001 |
| Percentage excess BMI loss | 1.02 (1.01–1.02) | 1.03 (1.02–1.03) | <0.001 |
| Age | 0.94 (0.93–0.95) | 0.94 (0.93–0.95) | <0.001 |
| Preoperative BMI | 1.06 (1.04–1.07) | 1.07 (1.05–1.09) | <0.001 |
| Sex | | | |
| Female | Reference | Reference | Reference |
| Male | 0.89 (0.77–1.02) | 1.57 (1.29–1.90) | <0.001 |
| Education | | | |
| Primary education (≤9 years) | 0.77 (0.65–0.92) | 0.96 (0.76–1.22) | 0.747 |
| Secondary education (10–12 years) | Reference | Reference | Reference |
| Higher education ≤3 years | 1.07 (0.86–1.34) | 1.15 (0.86–1.55) | 0.353 |
| Higher education >3 years | 1.27 (1.10–1.60) | 1.26 (0.94–1.70) | 0.126 |
| Surgical method | | | |
| Gastric bypass | Reference | Reference | Reference |
| Sleeve gastrectomy | 0.64 (0.47–0.88) | 0.72 (0.48–1.10) | 0.129 |

[1]Multivariable logistic regression including all factors listed in the table.

Although the mechanisms behind the metabolic effects of bariatric surgery are complex and incompletely understood, they appear to be mediated by a combination of weight-loss-dependent and -independent factors, resulting in improved insulin sensitivity and improved pancreatic beta-cell function [24–27]. Patients with longer duration of T2D are more likely to have reduced beta-cell function and secretory capacity. Although this group of patients will still benefit from the reduction in insulin resistance resulting from bariatric surgery, patients with reduced beta-cell function are less likely to achieve complete remission. This hypothesis is supported by the lower remission rates among patients with poorer glycemic control and insulin dependence prior to surgery, as seen in both the present study and previous studies [12–14]. Although higher preoperative HbA1c does not necessarily correlate to reduced beta-cell function, it indicates poorer glycemic control and greater severity of disease, factors known to reduce the chance of diabetes remission [12,23]. Preoperative insulin treatment, on the other hand, indicates significantly reduced beta-cell function that may not fully respond to the increase in incretin secretion after bariatric surgery. The strong negative correlation between preoperative insulin treatment and chance of achieving complete diabetes remission is well supported by reports from previous studies [12,15].

Age was found to be negatively associated with diabetes remission. Aung et al. also reported a negative correlation between age and the chance of diabetes remission [14]. In their study, patients with late onset of diabetes had a lower chance of durable diabetes remission after bariatric surgery [14]. Furthermore, older patients are known to lose less weight than younger patients after bariatric surgery [28], a factor also associated with reduced chance of achieving complete remission. A combination of a slightly different metabolic profile with poorer weight loss among older patients may well explain the negative association between age and diabetes remission. The influence of sex on diabetes remission is less clear. With the higher proportion of women in the study, it is possible that men with T2D represented a different subgroup than women with T2D due to different fat distribution. Indeed, the fat distribution pattern more

often seen in men is strongly related to metabolic complications of obesity [29]. The weight loss after bariatric surgery could thus have higher metabolic impact in patients with male fat distribution.

In the present study, higher preoperative BMI was associated with improved T2D remission rates in the univariable as well as the multivariable analyses. To our knowledge, this has not been reported in previous studies, possibly due to smaller patient numbers. Although no explicit explanation can be provided from the results of this study, it could be that T2D has different characteristics depending on the degree of obesity, where patients with a high BMI have a greater degree of insulin resistance. In patients with a lower BMI, genetic factors and reduced beta-cell function could play a greater role. Furthermore, patients with a high BMI tend to lose more weight after bariatric surgery [30], and thus experience a more significant weight-dependent metabolic effect of surgery.

Consistent with previous studies, gastric bypass appeared to be associated with higher rates of diabetes remission compared to sleeve gastrectomy in the unadjusted analysis [9,10]. Previous studies have reported similar effects on glucose homeostasis and incretin levels when comparing these 2 methods [31]. A plausible explanation may be the difference in postoperative weight loss seen in this and previous studies. However, the present study was not designed to compare the efficacy of the 2 methods, so this result should be viewed with caution. Further randomized trials comparing the effectiveness of the methods are still needed.

## Strengths and limitations

Despite the strengths of this study—i.e., the large number of patients, nationwide data, and the high quality of data, allowing estimation of treatment effects for bariatric surgical patients with T2D—there are limitations that must be acknowledged. The major limitation lies in the retrospective nature of the study. By using clinical registries, the study was limited to the variables and definitions already specified by those registries. Some valuable clinical variables, e.g., C-peptide and behavioral patterns, could thus not be included. Furthermore, diabetes remission was defined based on prescribed drugs. Despite reaching adequate glucose control, patients are sometimes encouraged by healthcare providers to continue certain diabetes drugs such as metformin [32]. This would give the impression that medication was still necessary, i.e., no remission, resulting in underestimation of the diabetes remission rate. It is unlikely, however, that this influenced the major outcomes of the present study. Finally, consistent with previous reports, high follow-up rates are difficult to achieve outside of randomized clinical trials [33]. In the present study, HbA1c values were only available for 27% of patients at the 5-year follow-up, limiting analysis of long-term complete remission. However, the patterns of pharmaceutical usage and complete remission were very similar at the 2-year and 5-year follow-ups after surgery, consistent with previous studies.

## Conclusion

Remission of T2D after bariatric surgery is negatively correlated to diabetes duration, with the highest rates among patients with more recent onset and less severe disease.

## Supporting information

**S1 STROBE checklist.**
(DOC)

## Author Contributions

**Conceptualization:** Ingmar Näslund, Eva Szabo, Erik Näslund, Erik Stenberg.

**Data curation:** Anders Jans, Ingmar Näslund, Johan Ottosson, Erik Stenberg.

**Formal analysis:** Anders Jans, Erik Stenberg.

**Funding acquisition:** Erik Näslund, Erik Stenberg.

**Investigation:** Anders Jans, Erik Stenberg.

**Methodology:** Ingmar Näslund, Erik Stenberg.

**Project administration:** Ingmar Näslund, Erik Näslund, Erik Stenberg.

**Resources:** Eva Szabo, Erik Näslund.

**Software:** Johan Ottosson.

**Supervision:** Erik Stenberg.

**Validation:** Ingmar Näslund.

**Visualization:** Ingmar Näslund, Erik Stenberg.

**Writing – original draft:** Anders Jans, Erik Stenberg.

**Writing – review & editing:** Ingmar Näslund, Johan Ottosson, Eva Szabo, Erik Näslund.

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
