## [Decision Letter · Decision Letter 0]

10 Sep 2019

Dear Dr. Stenberg,

Thank you very much for submitting your manuscript "The impact of duration of diabetes on remission rates after bariatric surgery" (PMEDICINE-D-19-02752) for consideration at PLOS Medicine. 

[LINK]

In light of these reviews, I am afraid that we will not be able to accept the manuscript for publication in the journal in its current form, but we would like to consider a revised version that addresses the reviewers' and editors' comments. Obviously we cannot make any decision about publication until we have seen the revised manuscript and your response, and we plan to seek re-review by one or more of the reviewers. 

We expect to receive your revised manuscript by Oct 01 2019 11:59PM. Please email us (plosmedicine@plos.org) if you have any questions or concerns.

We look forward to receiving your revised manuscript. 

Sincerely,

Adya Misra, 

Senior Editor 

PLOS Medicine

plosmedicine.org

Title, please recast to include a study descriptor. Perhaps:

Association with bariatric surgery and the duration of type 2 diabetes in Sweden 2007-2015: a retrospective cohort study

Abstract – this is structured with 3 sections: Background, Methods and Findings and Conclusions and please ensure the final sentence of the Methods and Findings section includes a sentence on the limitations of the study. 

Abstract – please include cities and some summary participant information such as age, sex, BMI, etc

Data – an author cannot be a point of contact – please find another contact point. 

References in the main text – please use square brackets and remove superscript

Was written consent provided by participants?

Did your study have a prospective protocol or analysis plan? Please state this (either way) early in the Methods section.

c) In either case, changes in the analysis—including those made in response to peer review comments—should be identified as such in the Methods section of the paper, with rationale.

Please provide a STROBE reporting guidelines (submitted as a Supp file and ensure paragraph and sections are used instead of page numbers – these will change in the event of publication.)

Comments from the reviewers:

Reviewer #1: I have read with great interest the paper by Jans et al. They have used the large Swedish registry data on bariatric surgeries, to answer the question that which factors contribute to remission of T2DM after surgically-induced weight loss. The study was well-designed and -written. I have only some very minor comments to it.

Abstract

At first read, it was slightly difficult to capture the concept of correlation in the sentence: "The chance of achieving complete remission correlated…" since there were both continuous (duration of diabetes, age, Hba1c) and dichotomous variables (insulin treatment) in the analyses. Can this be opened, e.g. by adding units after each OR.

Introduction

Obesity rates are low. Any data available for larger prevalences, which is the case in most countries?

Methods

Nro of patients in each operation type could belong to the methods, however if the authors wish, it can as well remain in the results section as it is now.

Results

A flow chart of the available patients would be useful. The percentages are hard to follow. For example in the sentence "5388 patients were available for analyses on pharmaceutical use at 5 years (98%)": What does the 98% represent?

Reviewer #2: Thanks for the chance to review this manuscript. This study verified that longer T2DM duration, insulin therapy prior to surgery, lower BMI were negative predictors to predict T2DM remission after RYGB/SG with a large sample size and longer follow-up period. Although these data from several registry systems was high-quality, some valuable clinical variables were missing, for instance, fasting C peptide. 

Many previous studies have found T2DM duration, C peptide, age, BMI and insulin usage were powerful predictive factors, and thus the novelty of this study is limited. Additionally, during follow-up period, most patients were lost, which affected the credibility of the conclusions.

Reviewer #3: Due to the imbalanced effects of bariatric surgery on clinically improving metabolic disorders among patients, exploring the factors that affect surgery-induced benefits is quite necessary. In this study, the author provides convincing evidences with large sample size to support his conclusion that the remission of T2DM induced by bariatric surgery is negatively correlated to duration of diabetes and increases in patient with recent onset diabetes and those without insulin treatment. These findings are meaningful for clinician to make decision on treating T2DM by surgery, a treatment that, although effective, cause irreversible changes in patient's GI track.

However, as the author mentioned in the manuscript, the retrospective property of this study limits its significance, other variables among subjects such as distinct behaviors during as long as 5 years may also contribute to the uneven diabetic remission rate after surgery, which need to be further and deeply explored. Whereas the high-quality data and large sample size of this study still make it convincible, and it should be eligible to be published in Plos medicine.

Reviewer #4: I confine my remarks to statistical aspects of this paper. In general, these were fine, although I do wonder why the authors did not do a Cox proportional hazards survival analysis.

Some comments/suggestions:

Line 127-8 I think it would be better to use "proportion of excess weight". I am guessing that the people in the study varied a lot in how overweight they were. If a person is 100 kg overweight and another is 200 kg overweight and each loses 100 kg, it does not have the same effect. Maybe there is no good estimate of ideal weight, but even BMI (flawed though that is) could be used.

Table 1 - What about exactly 10 years of education? or exactly3 years of higher education?

Figures - stacked histograms are not a good method (see the work of William S. Cleveland). Line charts would be better with duration on the x axis, percent on the y axis and a line for each outcome.

Peter Flom

[LINK]

---

## [Editor Report · Decision Letter 1]

23 Oct 2019

Dear Dr. Stenberg,

Thank you very much for re-submitting your manuscript "Association with duration of type 2 diabetes and remission rates after bariatric surgery in Sweden 2007-2015: a registry-based cohort study" (PMEDICINE-D-19-02752R1) for review by PLOS Medicine.

I have discussed the paper with my colleagues and the academic editor. I am pleased to say that provided the remaining editorial and production issues are dealt with we are planning to accept the paper for publication in the journal.

[LINK]

We look forward to receiving the revised manuscript by Oct 30 2019 11:59PM. 

Sincerely,

Adya Misra, PhD

Senior Editor 

PLOS Medicine

plosmedicine.org

Requests from Editors:

Please remove "association with" from the start of the title

Abstract “other important factors” – please be specific in text. In this case, specify or remove; 

Author summary – please remove ‘obesity surgery’ as I don’t think this is an official term, is it? 

- There are some "p<0.0001" in the abstract and results – please alter to <0.001, per house style

Line 221 – “4192 patients received oral treatment” please be specific, of what? And again line 226, please also correct anywhere else in the main text. 

- "In this study, we found that ... was ..." at line 58, or similar (and around line 88)

- I'd go for "negatively [or "inversely"] associated" rather than "negatively correlated"

- square brackets needed for refs in the main text

Comments from Reviewers:

[LINK]

---

## [Editor Report · Decision Letter 2]

31 Oct 2019

Dear Dr. Stenberg, 

On behalf of my colleagues and the academic editor, Dr. Kirsi H. Pietiläine, I am delighted to inform you that your manuscript entitled "Duration of type 2 diabetes and remission rates after bariatric surgery in Sweden 2007-2015: a registry-based cohort study" (PMEDICINE-D-19-02752R2) has been accepted for publication in PLOS Medicine. 

PRODUCTION PROCESS

PRESS

PROFILE INFORMATION

Thank you again for submitting the manuscript to PLOS Medicine. We look forward to publishing it. 

Best wishes, 

Adya Misra, PhD

Senior Editor 

PLOS Medicine

plosmedicine.org